Long-term resource addition to a detrital food web yields a pattern of responses more complex than pervasive bottom-up control

Lawrence Kendra L. 1
Wise David H. dhwise@uic.edu 2
1 KL2 Consulting, LLC , Silver Spring , MD , United States of America
2 Department of Biological Sciences, and Institute for Environmental Science and Policy, University of Illinois at Chicago , Chicago , IL , United States of America
Huber Dezene
Electronic publication date: 2017 Jul 21
Publication date: 2017
Volume: 5
Electronic Location ID: e3572
Received 2017 Jan 13; Accepted 2017 Jun 22
Copyright: ©2017 Lawrence and Wise
Copyright year: 2017
Copyright holder: Lawrence and Wise
License: This is an open access article distributed under the terms of the Creative Commons Attribution License, which permits unrestricted use, distribution, reproduction and adaptation in any medium and for any purpose provided that it is properly attributed. For attribution, the original author(s), title, publication source (PeerJ) and either DOI or URL of the article must be cited.
License URL: https://creativecommons.org/licenses/by/4.0/

Keywords: Leaf litter, Arthropod community, Field experiment, Control processes, Detritus, Long-term, Collembola, Araneae, Pseudoscorpiones, Coleoptera

Funding: US National Science Foundation DGE-9355093 DEB-9815842 DEB-0735236 Kentucky Agricultural Experiment Station Hatch Project KY008005 University of Illinois All support for this research came from US National Science Foundation Grants DGE-9355093, DEB-9815842 and DEB-0735236; Kentucky Agricultural Experiment Station Hatch Project KY008005; and resources and a sabbatical leave provided to DHW by the University of Illinois at Chicago for data analysis and manuscript preparation. There was no additional external funding received for this study. The publishing fee for this article was paid for by the Research Open Access Article Publishing (ROAAP) Fund of the University of Illinois at Chicago. The funders had no role in study design, data collection and analysis, decision to publish, or preparation of the manuscript.

==============================
Background

Theory predicts strong bottom-up control in detritus-based food webs, yet field experiments with detritus-based terrestrial systems have uncovered contradictory evidence regarding the strength and pervasiveness of bottom-up control processes. Two factors likely leading to contradictory results are experiment duration, which influences exposure to temporal variation in abiotic factors such as rainfall and affects the likelihood of detecting approach to a new equilibrium; and openness of the experimental units to immigration and emigration. To investigate the contribution of these two factors, we conducted a long-term experiment with open and fenced plots in the forest that was the site of an earlier, short-term experiment (3.5 months) with open plots (Chen & Wise, 1999) that produced evidence of strong bottom-up control for 14 taxonomic groupings of primary consumers of fungi and detritus (microbi-detritivores) and their predators.

Methods

We added artificial high-quality detritus to ten 2 × 2-m forest-floor plots at bi-weekly intervals from April through September in three consecutive years (Supplemented treatment). Ten comparable Ambient plots were controls. Half of the Supplemented and Ambient plots were enclosed by metal fencing.

Results

Arthropod community structure (based upon 18 response variables) diverged over time between Supplemented and Ambient treatments, with no effect of Fencing on the multivariate response pattern. Fencing possibly influenced only ca. 30% of the subsequent univariate analyses. Multi- and univariate analyses revealed bottom-up control during Year 1 of some, but not all, microbi-detritivores and predators. During the following two years the pattern of responses became more complex than that observed by Chen & Wise (1999). Some taxa showed consistent bottom-up control whereas others did not. Variation across years could not be explained completely by differences in rainfall because some taxa exhibited negative, not positive, responses to detrital supplementation.

Discussion

Our 3-year experiment did not confirm the conclusion of strong, pervasive bottom-up control of both microbi-detritivores and predators reported by Chen & Wise (1999). Our longer-term experiment revealed a more complex pattern of responses, a pattern much closer to the range of outcomes reported in the literature for many short-term experiments. Much of the variation in responses across studies likely reflects variation in abiotic and biotic factors and the quality of added detritus. Nevertheless, it is also possible that long-term resource enhancement can drive a community towards a new equilibrium state that differs from what would have been predicted from the initial short-term responses exhibited by primary and secondary consumers.

Introduction

Classical theory predicts extensive bottom-up control in detritus-based food webs. However, accumulating empirical findings and modeling research suggest that bottom-up control may not be as strong and pervasive as hypothesized, and that a mixture of bottom-up and top-down control processes characterizes soil food webs (e.g., Bardgett & Wardle, 2010; McCann, 2012; Moore & DeRuiter, 2012; Pimm, 2002; Wardle, 2002). Debates over the strength of bottom-up and top-down control in food webs with many generalist predators (a characteristic of soil communities) intensified in parallel with increasing reliance on controlled field experiments to test theory (e.g., Hairston, 1989; Hairston Jr & Hairston Sr, 1993; McCann, 2012; Menge, 2000; Moore et al., 2004; Moore & DeRuiter, 2012; Polis & Strong, 1996; Polis & Winemiller, 1996; Resetarits & Bernardo, 1998).

Making inferences about population control processes from field experiments is not straightforward (Raffaelli & Moller, 2000). Two major challenges are space and time. Species interactions occur over a spectrum of spatial scales. Finding an appropriate size for experimental units is difficult, especially given opposing constraints imposed by the desire for realism, the requirement of sufficient replication for adequate statistical power, and the frequent need to impose barriers to migration by using cages or fencing (Gardner et al., 2004; Hurlbert, 1984; Raffaelli & Moller, 2000; Resetarits & Bernardo, 1998). Perhaps an even greater challenge is time. Short-term experiments often fail to capture the spectrum of responses caused by temporal variation in abiotic factors such as rainfall. Furthermore, short-term experiments are unlikely to reveal new equilibria because detecting the consequences of time lags within chains of indirect effects requires many generations of interacting organisms (Bender, Case & Gilpin, 1984; Osenberg & Mittelbach, 1996; Yodzis, 1988).

The most direct way to uncover the strength of bottom-up control and its pervasiveness across trophic levels is to observe how adding energy- and/or nutrient-rich detritus to replicated plots in a field experiment alters densities of major taxa of primary and secondary consumers. Such experiments in terrestrial ecosystems have been conducted in prairies and grasslands (Fountain et al., 2008; Hoekman et al., 2011; Oelbermann, Langel & Scheu, 2008; Patrick, Kershner & Fraser, 2012) and forests (Chen & Wise, 1999; David et al., 1991; Lessard et al., 2011; Maraun et al., 2001; Raub et al., 2014; Salamon et al., 2006; Scheu & Schaefer, 1998; Yang, 2006). Most experiments uncovered some evidence for bottom-up control, but the strength and pervasiveness across trophic levels of the responses varied substantially. In some studies, supplementing detrital input resulted in increased populations of primary consumers and their predators. In other experiments the predators showed no response. Sometimes effects occurred across many taxa, sometimes they were limited to a few, and in a few instances effects of detrital supplementation were negative. Such variability in outcomes likely reflects differences between experiments in many factors, both environmental and logistical: levels of limiting abiotic factors (i.e., rainfall and temperature) during the experiment; type, quality, and amount of added detritus; taxonomic resolution of the response variables; plot size; number of replicates; whether the plots were open or fenced; and duration of the experiment.

Long-term experiments are more likely to capture the influence of variation in abiotic factors and also are more likely to reveal indirect effects that propagate at different rates through a complex food web. Among experiments conducted to date, densities of some trophic groups occasionally responded negatively to addition of detritus. Do these negative effects reflect differential responses by predators to resource-addition combined with intensification of top-down control processes along a subset of trophic pathways due to the indirect effects of trophic-level omnivory in a detritus-based food web (Polis & Strong, 1996; Halaj & Wise, 2002; Oelbermann, Langel & Scheu, 2008)? Were different pathways of top-down control processes modulating the strength of bottom-up control, producing different responses among trophic levels as the system moved towards a new equilibrium state? Direct manipulation of predator densities in similar detritus-based food webs has revealed top-down control, including trophic cascades affecting decomposition rates (Kajak, 1997; Kajak & Jakubczyk, 1977; Lawrence & Wise, 2004; Lensing & Wise, 2006; Wardle, 2002; Wise, 2004). However, negative effects of detrital enhancement on densities of primary consumers and predators were only infrequently observed in previous experiments. A caveat is needed, though, as most resource-enhancement experiments have been relatively brief compared to the generation times of species in the community (one exception is the 4-year manipulation of nutrients in a grassland (Patrick, Kershner & Fraser, 2012)). Furthermore, no experiments of both short-and long-term duration have been conducted with the same system, so the question remains unanswered of whether short-term resource enhancement produces patterns predictive of longer-term resource-enhancement experiments.

Detrital-enhancement experiments have been conducted with both fenced and open plots. Fencing, which has the potential to reduce confounding effects of emigration and immigration on responses of the target taxa to resource enhancement, is potentially most critical for smaller experimental units. This is especially true if the experiment is conducted over many generations of the affected organisms. Positive responses in open plots possibly underestimate the actual strength of bottom-up control because populations that increased in response to detrital supplementation can decline as emigrants leave for areas outside the plots where densities are lower. Similarly, unexpected decreases in density in supplemented unfenced plots might be masked by augmentation of densities due to immigration from outside. Thus, we hypothesized that fencing the plots would produce more realistic estimates of the strength of bottom-up control, or conversely, would more likely reveal population declines resulting from cascading indirect effects.

Here we report results of a 3-year detrital-addition experiment in a secondary oak-maple-hickory forest, with a few scattered pine trees, in Madison Co., Kentucky, USA. Our experiment was conducted within ∼0.5 km of the sites of a previous similar, but short-term (3.5-months), experiment that utilized open (unfenced) forest-floor plots (Chen & Wise, 1999)—referred to as “CW99” from now on. The earlier experiment in this forest produced clear evidence of detrital resource limitation (sensu Osenberg & Mittelbach (1996)) i.e., relatively rapid elevation of population densities of both primary and secondary consumers in response to addition of high-quality detritus. Major groups of primary consumers, the microbi-detritivores (grazers of fungi and consumers of organic debris), were at least 2–3× more abundant in supplemented than control plots. Major arthropod predators were ∼2× as abundant. No taxa were less abundant in the resource-supplemented treatment. Because the plots were not fenced, emigration of some taxa that responded positively to resource supplementation might have weakened the observed magnitude of bottom-up control. Similarly, immigration of taxa that had actually declined in density might have erased the negative treatment response. Our experiment expanded the design of CW99 to answer two questions (the first, logistical; the second, conceptual): (1) How would a barrier that reduced emigration/immigration of some groups of ground arthropods affect conclusions about bottom-up control? (2) Would a longer-term experiment reveal inconsistences in responses that would suggest either tempering of bottom-up control by variation in abiotic factors and/or movement of the perturbed community in the direction of a new equilibrium?

Methods

Experimental design

The experiment started two years after CW99 and ran from 1997 through 1999 (hereon designated Years 1, 2, and 3). Each experimental unit (20 in total) was a 2 × 2-m area of forest floor, separated from each other by at least 10 m. Experimental units were randomly assigned to one of two levels of a resource treatment and one of two levels of a fencing treatment, yielding five replicates of each of the four combinations of treatment levels. Thus, half of the units received a detrital supplement (Supplemented), the others none (Ambient), and half of the plots in each resource treatment were open to emigration and immigration (Open), while the others were enclosed with 35-cm aluminum flashing inserted 8 cm into the ground (Fenced). The fence was topped with a 15-cm horizontal strip of flashing that formed two lips coated on the underside with a tree-banding compound (Tanglefoot®, Grand Rapids, MI, USA) to further retard movement of epigeic (ground-active) arthropods across the barrier. This design is more complex than that of CW99, which had no fenced plots but employed the same total number of experimental units: twenty 2 × 5-m open plots, half of which received a detrital supplement.

We employed the detritus-supplementation protocol of CW99, which has also been used in other experiments (Chen & Wise, 1997; Raub et al., 2014). Our goal was not to determine which components of the resource base (bacteria, fungi, and organic debris) were possibly limiting densities of microbivores and detritivores, as was the aim of Salamon et al. (2006). Rather, our goal was to follow the community response to long-term enhancement of a broadly defined resource base. This experimental approach, including that of Salamon et al. (2006) and others, involves adding artificial forms of organic matter and nutrients. Every two weeks from April through September we added chopped “fresh” (i.e., not dried) mushrooms and potatoes, and dry flakes of Drosophila medium (Carolina Biological Supply; Burlington, North Carolina, USA; Formula 4–24) to the Supplemented plots.

We decided initially to supplement at a rate approximately 1/3 that of CW99 because we hypothesized that the strong responses exhibited in the earlier experiment were due to a high level of detrital enhancement. We planned to continue this rate of supplementation in the following years, but decided to increase the rate because the increase in densities of most taxa in response to the detrital enhancement in Year 1 was much less (including no responses) than that observed by CW99. Therefore, in Years 2 and 3 we increased the rate of supplementation to a level similar to CW99. In Year 1 each Supplemented plot received 195 g (dry wt.) m−2 of detritus (26 g m−2, 79 g m−2 and 90 g m−2 of mushrooms, potatoes, and Drosophila medium, respectively). In Years 2 and 3 the rate of detritus supplementation was increased ca. 4×  (to 770 g m−2 and 874 g m−2 total dry wt., respectively). The slightly larger amount of detritus added during Year 3 reflects a slightly longer period of detrital addition than in Year 2. Biweekly rates were the same in Years 2 and 3. This change in the rate of resource addition after Year 1 meant that statistical analyses focusing on the Resource × Time interaction only included Years 2 and 3, when rates of resource addition were the same.

Adding artificial detritus can influence the structure of the leaf-litter layer, so it is instructive to compare amounts added with litter standing crops. In Year 1 the average (±SE) standing crop of litter (dry wt. m−2) in Ambient plots was 671 ± 43 g (n = 10). Thus, the dry weight of detritus added throughout Year 1 was ∼30% of the average standing crop of detritus, compared to an addition rate of ∼100% of the litter standing crop in CW99. In Years 2 and 3 each Supplemented plot received ∼136% and ∼122% of that year’s average standing crop of litter, respectively. By the end of Year 3 litter weight was only 13% higher in Supplemented than Ambient plots (F1,16 = 4.93, P = .04), compared to a ∼30% increase at the end of CW99.

Due to our decision to increase the supplementation rate in Years 2 and 3 to ∼4× the rate of Year 1, differences in response patterns between Year 1 and the following two years can be attributed not only to differences in abiotic factors between years and time lags in the appearance of direct and indirect effects, but also to markedly different rates of detrital supplementation. In contrast, differences between Years 2 and 3 must have been due primarily to factors other than a difference in the rate of resource addition, since biweekly rates of supplementation in Years 2 and 3 were the same.

Fungal biomass

Differences in fungal abundance between Supplemented and Ambient plots at the end of the experiment were estimated by assaying leaf litter for ergosterol (Appendix S1), a common sterol in fungal hyphae that is nearly absent from plants (Weete & Weber, 1980). The amount of ergosterol is correlated with both total hyphal mass and membrane content and likely assayed both living and dead hyphae (Mille-Lindblom, Wachenfeldt & Tranvik, 2004; Ruzicka et al., 2000; Zhao, Lin & Brookes, 2005).

Arthropods

Selected taxa of arthropods active within and on the surface of leaf litter were sampled three times in Year 1 (18 July, 15 August, 13 October), twice in Year 2 (20 June, 23 September) and twice in Year 3 (3 June, 5 September). Values for the first two dates in Year 1 were averaged, yielding two values (Summer and Fall) for each year. Densities of most taxa were estimated by litter extraction and/or litter sifting. Sticky traps were used to measure activity-densities of adult flies (Diptera) just above the litter layer.

Litter extraction

Densities (measured as number per .05-m2 of forest floor) of 15 taxonomic groupings (six families of springtails (Collembola), thrips (Thysanoptera), larval moths (Lepidoptera), larval and adult beetles (Coleoptera), larval and adult flies (Diptera), pseudoscorpions (Pseudoscorpiones), centipedes (Chilopoda) and spiders (Araneae)) were estimated by taking litter from .05-m2 of forest floor per plot and using a temperature/humidity gradient in a modified Kempson-McFadyen apparatus (Kempson, Lloyd & Ghelardi, 1963; Schauermann, 1982) to extract animals into 50% ethylene glycol over 10 days. Extracted arthropods were washed and stored in 70% ethyl alcohol until identified.

Litter sifting

Litter extraction is not the best way to sample densities of active, larger spiders that often are less abundant than smaller life stages and species (Chen & Wise, 1999). Therefore, densities of cursorial spiders (primarily Corinnidae, Clubionidae, Gnaphosidae, and Lycosidae) and web-weaving spiders (primarily Linyphiidae and Dictynidae) were assessed by carefully sifting and searching, in the field, one 0.2-m2 sample of litter per plot (different from the .05- m2 sample).

Sticky traps

Densities (number per .05-m2) of larval and adult Diptera within the litter were determined by Kempson extractions. In addition, activity-densities (combined result of density and activity; number captured per trap) of selected families of adult Diptera were assessed with aerial traps placed just above the litter layer. Two 10 × 10-cm vertical pieces of metal insect screening attached to thin steel rods were coated with Tanglefoot® and placed 0.5m apart and perpendicular to each other in each plot for 24 h.

Taxonomic resolution of response variables

Several criteria dictated the taxonomic resolution of the response variables: (1) a close-as-feasible match to the degree of resolution reported in CW99; (2) logistical constraints (e.g., time and technical help) related to sorting and identifying the number of arthropods collected; (3) the overall pattern of relative abundance (i.e., numbers of each family of Collembola were greater than numbers for each of the other categories, all of which were orders or selected families within orders); and (4) the fact that trophic position (i.e., predator or detritivore) was broadly similar for organisms within orders, with the exception of Coleoptera (CW99 reported total Coleoptera and Staphylinidae, but did not distinguish other families; results were similar in CW99 for total Coleoptera and Staphylinidae)). Diptera in the litter were fungivorous families. Only fungivorous familes (Mycetophilidae, Sciaridae, and Phoridae) are included in the Diptera numbers from the sticky traps. Note that Diptera adults and spiders each are represented in Kempson samples and also in a different response variable, either sticky traps (Diptera) or litter-sifting (cursorial and web-building spiders). Cursorial and web-building families utilize very different foraging modes (Wise, 1993) and could easily be distinguished in sifting samples. They were pooled for Kempson samples due to the challenge of correctly identifying small specimens to family. Logistical constraints forced us to exclude mites (Acarina), a group that is abundant in leaf litter and was sampled in CW99.

Statistical analyses

Effects of resource addition and fencing were first analyzed for Year 1 to determine how the arthropod community responded to the lower rate of detrital supplementation. We then analyzed responses for Years 2 and 3 using a repeated-measures 2 × 2 × 2 design (Resource (Ambient, Supplemented)  ×  Fencing (Open, Fenced)  ×  Year (2 and 3)). All statistical modeling was done separately for summer and fall samples (rationale is explained below). The approach just described was used for both multivariate and univariate analyses. Methodological details of the statistical modeling appear in Appendix S2.

The central conceptual question addressed by our field experiment is: How does the response of the system to resource supplementation change over time? The statistic that directly addresses this question is the Resource  × Year interaction that includes only Years 2 and 3, when the rate of resource input did not vary between years. We evaluated this interaction by relying on a combination of multivariate and univariate statistical techniques. We first used multivariate techniques to determine how detrital supplementation changed arthropod community structure over time and which response variables were most closely linked to these changes. Multivariate analyses were performed first because if there is no multivariate effect, there is no justification for doing separate univariate analyses. Because multivariate effects were present we then conducted independent univariate analyses. This same approach was use to analyze the simple effect of resource supplementation in Year 1.

We postulated that the system would respond differently in summer and fall because summer samples had been exposed to detrital enhancement for fewer months than fall samples and because life histories of many taxa show pronounced seasonal patterns. Initial multivariate analyses of the interaction between (Resource  × Year) and Season confirmed this expectation (Tables S3.1, S3.2 and S4.1 in Appendices S3 and S4). Thus, we divided the community distance matrix into summer and fall subsets, yielding one distance matrix for each season. All subsequent multivariate and univariate analyses were done separately for summer and fall samples.

The central logistical question addressed by our field experiment is: How does the openness of the plots affect the observed pattern of responses to detrital supplementation? Thus, we first tested for an interaction between (Resource  × Year) and Fencing. If there was no evidence for an interaction (P value of the 3-way interaction > .10), the design was collapsed (Open and Fenced plots were pooled, yielding twice as many replicates per Resource level) to one involving only Resource and Year. If Fencing possibly influenced the Resource  × Year interaction, further analyses were done separately for Open and Fenced plots. Use of P > .10 (instead of the conventional P > .05 criterion) for inferring no interaction with Fencing made it less likely to ignore a weak effect of fencing (due to the relatively low statistical power of only five replicates for each Resource/Fencing combination) that could have influenced interpretation of the response patterns.

Multivariate analyses

Using all 18 response variables, we calculated a distance matrix using Gower’s similarity index (S15 of Legendre & Legendre (2012)). Gower’s index was employed because it is designed to accommodate different types of variables (Kempson, sifting, and sticky-trap samples; Fig. 1). We implemented S15 because this version of Gower’s measure is a symmetrical index that gives equal weight to double zeroes (absence/absence) and ++ (presence/presence), which is the type of distance measure philosophically appropriate for analyzing results of a field experiment (unlike the more commonly employed “Bray-Curtis” measure, which is less suited to the assumptions behind controlled manipulative experiments Legendre & Legendre (2012)). Further details are in Appendix S2.

Figure 1 N (ordinate) = total number extracted, encountered in litter sifting, or trapped during the experiment.

Response variables (combinations of taxa and sampling methods) pooled across treatments, seasons. and years. (A) All response variables (N = 18); (B) Non-Collembola response variables (N = 12). Variables are arranged in descending order of abundance. Key to abbreviations is based upon the three different sampling techniques. KEMPSON: Collembola (springtails)—Hyp, Hypogastruridae; Ony, Onychiuridae; Ent, Entomobryidae; Iso, Isotomidae; Tom, Tomoceridae; Smi, Sminthuridae; Thy, Thysanoptera (thrips); Llep, larval Lepidoptera (moths); Lcol, larval Coleoptera (beetles); Ara, Araneae (total spiders); Ldip, larval Diptera (flies); Pse, Pseudoscorpiones; Acol, adult Coleoptera (beetles); Adip, adult Diptera (flies); Chi, Chilopoda (centipedes). LITTER SIFTING: Cur, cursorial spider families (primarily Corinnidae, Clubionidae, Gnaphosidae and Lycosidae); Web, web-weaving spider families (primarily Linyphiidae and Dictynidae). STICKY TRAPS: TrpDip, adult Diptera (flies). Note that estimates of spider densities and adult Diptera numbers (density and activity-density) come from two different sampling methods.

Permutational multivariate analysis of variance (perMANOVA) (Anderson, Gorley & Clarke, 2008) was then used to assess the multivariate Resource × Year interaction. Because our analyses revealed a Resource × Year interaction, we then assessed the changing impact of Resource on arthropod community structure in two ways. First, we plotted the location of each experimental unit (i.e., 2 × 2-m plot) on the first two axes of the Principal Coordinates Ordination (PCO) for each year along with the P value for the Resource effect each year. Secondly, we evaluated how different response variables contributed to the change in community structure by plotting both simple and multiple correlation vectors on constrained PCO ordinations (CAP; constrained by Resource and Fencing treatments) for each year.

Univariate analyses

We plotted yearly means ± SE of each response variable and then used a range of statistical modelling approaches to help evaluate the change over time in differences between Supplemented and Ambient treatments. Details appear in Appendix S2.

Overall interpretation of response patterns

Interpretation of how the system responded was accomplished by evaluating the results of multivariate and univariate analyses taken together as a whole. Emphasis was placed upon using the statistical analyses to aid in interpreting the overall patterns of the ordinations and univariate plots. We avoided a completely NHST (Null Hypothesis Significance Testing) approach as much as possible. Thus, we made no corrections for “multiple comparisons” in the univariate analyses, primarily because multivariate effects were clear. Furthermore, we did not conclude that P values close to .05 but slightly greater was evidence for the absence of a response (Appendix S2). Instead, we relied on P values as a measure of strength of evidence (Cumming, 2012; Hector, 2015; Nakagawa & Cuthill, 2007). Our overall evaluation of the impact of resource enhancement on the suite of response variables was a melding of (1) changes in community structure as indicated by perMANOVA and PCO, (2) patterns of correlation-vector overlays on constrained ordinations (CAP), (3) univariate effect sizes estimated from plots of numbers for Resource and Ambient treatments over time, and (4) the P value of the appropriate statistic—simple Resource effect for Year 1; and for Years 2 and 3, the Resource × Year interaction, or the overall main effect of Resource if there was no change between Years 2 and 3 in the Resource effect.

Results

Fungal density

Fungal hyphae, as measured by concentrations of ergosterol, were 3× denser in Supplemented than in Ambient plots (Appendix S1).

Relative abundance of taxa

The most abundant taxa were six families of microbi-detritivores (Collembola) collected by Kempson extraction, with the Hypogastruridae, Onychiuridae, and Entomobryidae being the three most numerous (Fig. 1A). Remaining response variables were similar in value to each other, differing by less than 50%. Centipedes (Chilopoda) were the exception. This predatory taxon had the fewest individuals sampled of all response variables (Fig. 1B).

Response of arthropods to detrital supplementation

Results of multivariate analyses are presented first because they constituted the critical first step in the analysis. Results of testing for interactions with Fencing are presented before summarizing evidence for changes in the presence and strength of bottom-up control processes over the three years of the experiment. In each section below, results for Year 1 are presented first, followed by the pattern of change between Years 2 and 3, when the rate of Resource supplementation was the same and was ∼4× higher than in Year 1.

Multivariate modeling, including univariate vector overlays

Fencing — Permutational multivariate analysis of variance (perMANOVA) uncovered no evidence of an interaction between the openness of a plot and the overall community response to the addition of detritus for either summer or fall (Appendix S3) over the course of the experiment. Therefore, Open and Fenced plots were pooled for multivariate analyses of the Resource effect in Year 1 and the Resource × Year interaction for Years 2 and 3.

Addition of detritus — In the summer of Year 1 there was no clear divergence in arthropod community structure between Ambient and Supplemented plots (Fig. 2A, Appendix S4). By fall the centroids appeared to have diverged slightly more between treatments, but evidence for a pronounced Resource effect was not strong (P = .035, Fig. 2B, Appendix S4).

Figure 2 Changing impact over time of detritus addition upon arthropod community structure.

Principal Coordinates Ordinations (PCO on Gower’s distance measure; Appendix S2) presented separately for (A) SUMMER and (B) FALL. Open and Fenced plots have been pooled because there were no interactions involving (Resource × Year) and Fencing (Tables S3.1 and S3.2 in Appendix S3). P-values for the Resource treatment are given for each Year. Biweekly rates of detritus addition were the same in Years 2 and 3 and were ∼4 × higher than in Year 1. The effect of Resource addition on community structure varied between Years 2 and 3 (P(Resource × Year) = .012, Table S3.2 in Appendix S3).

Figure 3 Constrained ordinations (CAP; Resource and Fencing are the constraining factors) with vector overlays representing correlations between response variables and the two axes (Appendix S2).

Separation of communities along Axis 1 is largely related to the impact of the Resource treatment on community structure. Thus, the extent to which a vector is parallel with Axis 1 reflects the extent of the negative (to the left) or positive (to the right) correlation of densities of that response variable with the addition of detritus. The length of each vector represents the joint correlation of the response variable with both axes of the ordination, with the circle representing a correlation of 1. To prevent clutter on the graph, arrow heads of the vectors are not drawn. Key to abbreviations is in Fig. 1. (A) Vectors represent Spearman rank correlation coefficients with CAP Axis 1 that are ≥.50 or ≤−.50. (B) Vectors represent multiple correlation coefficients (analogous to univariate partial correlation coefficients) that are ≥.35 or ≤−.35.

In contrast, divergence in community structure due to the addition of detritus was clear for both seasons in Years 2 and 3, as indicated by the small degree of overlap between communities in Ambient and Resource treatments in ordination space (Fig. 2, Appendix S4). The impact of the Resource treatment on arthropod community structure in Year 3 differed from that of Year 2 (P(Pseudo-F1,70(Resource × Year) = .012, perMANOVA; Table S3.2, Appendix S3). Lower dispersion among Ambient plots in the fall of Year 3 (Fig. 2B) is likely causing the Resource × Year interaction; this is the only time when within-treatment dispersions in ordination space differed (Fall, Year 3: Pseudo- F1,18 = 8.85, P = .007, PERMDISP test (Anderson, Gorley & Clarke, 2008)).

Comparing simple Spearman and multiple (“partial”) correlations between response variables and the two axes of a constrained ordination (CAP) gives some insight into which taxa may have been the primary drivers of the observed divergence in community structure (Fig. 3). The identity of the response variables most highly correlated with the divergence changed over time. The pattern of these correlations also differed between seasons. Furthermore, several correlations were negative, indicating the community divergence was not due solely to simple bottom-up control of the response variables.

Simple Spearman Correlation Coefficients

Summer — In the summer of Year 1 the plots were not clearly separated according to detritus treatment, so the vector for isotomid Collembola does not reflect effects of detritus addition (Fig. 3A). In the following two summers, vector patterns reveal bottom-up control of several primary consumers that only weakly reached the secondary-consumer level (Fig. 3A). All response variables with coefficients ≥.50 were positively associated with the first axis, which is strongly linked to the detritus-addition treatment. Seven of the ten vectors for Years 2 and 3 represent microbi-detritivores (three for Diptera and four for Collembola (Hypogastruridae, Sminthuridae, and Entomobryidae)). The remaining three vectors represent simple correlations with larval and adult Coleoptera, which include both predators and microbi-detritivores.

Fall — The pattern for fall samples is more complex (Fig. 3A). Years 1 and 2 exhibited a pattern broadly similar to that of summer samples for Years 2 and 3, although the total number of responding taxa was greater in fall than summer (16 vectors versus 10, respectively). All vectors were positively correlated with CAP Axis 1 except for Thysanoptera, a consumer of detritus/fungi. Among the other microbi-detritivores, all six Collembola families showed increased densities in the Supplemented treatment in the fall of Year 1 and/or Year 2, as did larval and adult Diptera. More predatory groups had responded to detritus addition in the fall than in the summer. Vectors for both cursorial and web-building spiders in litter-sifting samples were positively associated with the first axis in Year 1 but not Year 2. In Year 2, larval and adult Coleoptera, many of which are predators, responded positively to detritus addition. The pattern for Year 3 in fall samples is strikingly different. Only two microbi-detritivore groups displayed positive vectors, and one group, larval Lepidoptera, had a weak negative correlation with Axis 1. The strongest correlation in Year 3 was that for web-building spiders. However, the relationship with detrital addition was negative, not positive as was the case in Year 1 (Fig. 3A).

Multiple (Partial) Correlation Coefficients

The multiple (partial) coefficient removes correlations with other response variables (Anderson, Gorley & Clarke, 2008). Thus, it is not surprising that fewer show a relationship with Axis 1, even with a lower R2 threshold, than do vectors representing the simple Spearman statistic (11 vs. 31 vectors, respectively; Fig. 3B vs. Fig. 3A (ignoring Summer of Year 1, when arthropod community structure showed no clear response to adding detritus)). Among the nine multiple-correlation vectors exhibiting a component positively correlated with Axis 1, only two include some predatory taxa (larval and adult Coleoptera) (Fig. 3B). Among the four vectors that indicate a negative response to detrital supplementation, one is that of a microbi-detritivore (tomocerid Collembola). The other three negative vectors are strictly predatory taxa: pseudoscorpions and total spiders (Ara) in Kempson samples, and web-building spiders (Web) in litter-sifting samples (Fig. 3B).

Univariate statistical modeling

Plots over time of all 18 response variables appear in Figs. S5.1–S5.18 in Appendix S5. Full results of the statistical modeling of the univariate responses, including a comparison with the vector patterns in Fig. 3, appear in Tables S6.1–S6.4 in Appendix S6. These analyses yield a complicated pattern of responses that is best summarized pictorially (Fig. 4A; note that Fig. 4A also includes some symbols for vector overlays, which have been excluded from the tallies in the following sections but will be treated in the Discussion).

Figure 4 Temporal pattern of responses of all 18 response variables to supplementing the resource base, derived from plots of treatment means over time (Appendix S5), statistical modeling of univariate responses (Appendix S6), and vector overlays (Fig. 3).

Response variables in each category are listed according to descending overall values (Fig. 1). The symbol for the CAP vector only indicates the direction of the correlation; the width does not reflect the strength of the correlation with CAP Axis 1. The symbol for the CAP vector only appears if the univariate model failed to reveal an effect (P > .10). If a vector appears in Fig. 3 for a response variable for which the univariate statistical modeling revealed an effect, only the arrow summarizing the univariate model is presented. The width of arrows summarizing effect size for the univariate model reflects the ratio of Supplemented to Ambient mean values (Appendix S5). The asterisks indicate the P value for the appropriate term in the univariate statistical model associated with the effect size (Appendix S6) ((∗) = P ≤ .10; ∗ = P ≤ .05; ∗∗ = P ≤ .01; ∗∗∗ = P ≤ .001). Effects with P > .10 are not indicated, with just one exception (TrpDip, Summer Years 2 and 3; see Table S6.2 in Appendix S6). If there was an interaction with Fencing, the arrow describes the fencing treatment that displayed a simple Resource effect or a Resource × Year interaction, indicated by “O” or “F” to the right of the arrow for Open or Fenced plots, respectively. Arrows without a letter depict analyses based upon pooled Open and Fenced plots (N = 10 / Resource treatment). Arrows and asterisks for Year 1 summarize the simple Resource effect that year. The pattern for Years 2 and 3 is more complex. An arrow for one year and a blank cell for the other year indicates a Resource × Year interaction for that season. The asterisk refers to the P value of the Resource × Year interaction. If there is no Resource × Year interaction but there is an overall Resource effect in the model, an arrow is present for each date and the asterisks reflect the P value of the overall Resource effect. (B) Results for the study by CW99, which are based upon 3.5 months of adding detritus to open (unfenced) 2 × 5-m plots (N = 10 / Resource treatment) (Chen & Wise, 1999). N/A = response variable not sampled.

Fencing — Although the multivariate analyses uncovered no interactions with Fencing, ∼30% of the univariate tests that revealed an effect of Resource (10/34; only Chilopoda, the rarest taxon, never responded) exhibited an interaction with the Fencing treatment (Fig. 4A, Appendix S6). For most groups the effect was sporadic, i.e., it occurred in only one of the six sampling periods. The two clear exceptions were larval Lepidoptera and pseudoscorpions (Fig. 4A). In both groups an interaction with fencing occurred in three or four sampling periods and always was associated with a decrease in numbers in response to detrital supplementation. The negative Resource effect occurred four times in Open plots, three times in Fenced plots

Addition of detritus — The pattern of response changed over the three years (Fig. 4A). In Year 1 eleven response variables showed a positive response to detrital addition in at least one season; the only exception was the lower number of Lepidoptera larvae in Open plots in the summer (arrows for vector overlays are not included in this and following tallies). In Year 2 eleven response variables also showed a positive response in one or two seasons, but only six of these had displayed a positive response the earlier year. In Year 3 only five variables showed a positive response in summer and/or fall (all had exhibited a positive response the previous year). In Year 2 three variables decreased in response to detrital addition in at least one season and in Year 3 four taxa responded negatively.

Thus, evidence of predominantly bottom-up control (i.e., a positive response to adding detritus) by the fall of Year 1 was transformed the following two years into a mixture of continued positive responses to detrital addition, disappearance of the early positive responses, appearance of new positive responses, and appearance of negative effects of adding detritus. Individual patterns are presented below by trophic categories.

Fungivores / Detritivores (microbi-detritivores)

Five of the six Collembola families (first entries of Fig. 4A) displayed evidence of strong bottom-up control, i.e., an effect size of  2× to >3× in at least two seasons over the three years, with the highest number of strong responses in Year 2. Entomobryidae exhibited persistently strong bottom-up control throughout the experiment, with a strong response (>3×) each fall and weaker but positive responses to detrital addition each summer. The most extreme pattern is that of tomocerid Collembola: a positive response in Fenced plots the fall of Year 1, no response in Year 2, and a negative response in the summer of Year 3. By the fall of Year 3 densities of onychurids, isotomids, tomocerids and sminthurids were close to zero in all plots (Figs. S5.2, S5.4–S5.6 in Appendix S6).

Diptera (trapped adults and larvae and adults from Kempson samples) showed positive responses to Resource addition, though responses were sporadic. Larval Diptera displayed the most consistently strong response, particularly in the fall of the last two years. Thysanoptera also responded positively, but only in the first two years.

In marked contrast to the above patterns, larval Lepidoptera displayed a negative, not positive, response to detrital enhancement in five of six sampling periods, but the effect size usually was small and was influenced sporadically by openness of the plots.

Mixed Trophic Levels

Larval Coleoptera exhibited higher densities in the Supplemented treatment in Years 2 and 3 but did not respond to the detrital treatment in Year 1. Adult Coleoptera showed a clear response to detrital enhancement only in the fall of Year 2.

Predators

The Chilopoda (centipedes), the rarest taxon among those sampled (Fig. 1), was the only group to exhibit no response to detrital enhancement during the experiment (Fig. 4A, Appendix S6). The other strictly predatory taxa responded, but the direction of the response changed strikingly over the three years. Cursorial spiders in the litter-sifting samples showed a very weak positive response to Resource enhancement in the fall of Year 1, but no response until the fall of Year 3, when they were less abundant in unfenced Supplemented plots than comparable Ambient plots. Total spiders in the Kempson samples were more abundant in detrital-addition plots the first summer, but never differed between Resource treatments during the rest of the experiment. Pseudoscorpions showed no response to detrital enhancement in Year 1, and were less, not more, abundant in Open or Fenced Supplemented plots, compared to comparable Ambient plots, the final four sampling periods. Web-building spiders had exhibited a clear positive response to detrital enhancement by the fall of Year 1, but this response gradually disappeared over the experiment (Fig. S5.17, Appendix S5).

Raw Data

The data on which analyses are based can be accessed in Appendix S7, which contains a description of both response and design variables, and in Appendix S8, which contains the raw data and values of the design variables.

Discussion

Many ecologists have pointed to the strong positive response to detrital enhancement by both primary consumers and their predators observed by CW99 as evidence of pervasive, strong bottom-up control in terrestrial detritus-based food webs. According to Google Scholar, CW99 has been cited frequently—237 citations as of May 2017 (31 citations in 2015–2017). Our long-term experiment does not support this generalization, particularly with respect to the predators. Fourteen of our 18 response variables are shared with CW99 (Fig. 4). The mixture of outcomes for these shared response variables in our 3-year experiment contrasts markedly with the strong, bottom-up control observed for all 14 variables in the same forest just a few years earlier in the shorter-term (3.5 months) experiment (Fig. 4B vs. Fig. 4A). Only eight of the shared variables (five fungivores/detritivores (microbi-detritivores) and three predators) exhibited bottom-up control in Year 1 of our experiment (considering univariate results and vector overlays for both seasons) and effect sizes also tended to be smaller than in CW99—a pattern most likely a consequence of our having added detrital resources at 1/3 the rate of CW99. This inference is supported by the fact that in Year 2, when the rate of supplementation was increased 4× to a level slightly higher than that of CW99, all Collembola families except Tomoceridae, and both larval and adult Coleoptera, responded to additional detrital resources is ways very similar to CW99 (Fig. 4). However, this similarity for the lower trophic levels contrasts sharply with the failure of any strictly predatory group to show an increase in Year 2 of our experiment in response to increased detrital input, in marked contrast with CW99. What is perhaps most surprising about our findings is the complete disappearance of evidence of bottom-up control among predators in the final two years of our study. In fact, during Years 2 and 3 all responses by predator populations to an increased basal resource were negative, not positive. In addition, the overall difference in responses between Year 3 and both Year 2 and CW99 is also striking, especially since rates of detrital supplementation were similar across years.

First, we will evaluate possible explanations for why our results are more variable than those of CW99. Then we will weave together our findings and those of CW99 with the variable evidence for bottom-up control in terrestrial food webs revealed by similar field experiments. It will become clear that variation in outcomes reported to date most likely reflects differences between studies in uncontrolled abiotic and biotic environmental factors, amount and nature of the added detrital resource, taxonomic resolution of the response variables, plot size and openness, and duration of the experiment. We will conclude by speculating about new equilibrium states and offering suggestions for future research.

Comparison with experiment of Chen & Wise (1999) (CW99)

Fencing and statistical power— Fencing half of our plots meant that we had 10 open plots compared to the 20 open plots of CW99. Thus, when the statistical model included an interaction with Fencing, the number of replicates per Resource treatment was only 5 instead of 10 as in CW99. This reduced statistical power, which occurred in ∼30% of the univariate models, would have made it more difficult to detect responses to detrital supplementation that were comparable in magnitude to CW99. However, in the remaining 70% of the univariate models the absence of an interaction with Fencing meant that the statistical power for detecting a Resource effect was comparable to that of CW99. The three instances of a fencing interaction for the microbi-detritivore taxa shared with CW99 (one sampling period each for Hypogastruridae, Tomoceridae and Smithuridae; Fig. 4A) confirmed our expectation that fencing can increase the evidence for bottom-up control, since the positive Resource effect occurred only in the Fenced plots. The barrier may have yielded a greater positive response by reducing emigration, since Collembola are not likely to climb the type of barrier we constructed. The interaction with fencing for Collembola, however, was relatively infrequent; and the direction of the interaction brought the overall pattern of Collembola responses in our experiment closer to those of CW99.

Among the primary consumers two other taxa also exhibited a similar interaction with Fencing on one sampling period (Thysanoptera and adult Diptera). Lepidoptera larvae, which were not sampled in CW99, showed a surprising pattern of weak negative responses across the entire experiment peppered with contrasting effects of Fencing (Fig. 4A). The appearance of a negative response at the very start of the experiment, and the fact that no other response variable exhibited a negative response in Year 1, suggests that the pattern is due to a sampling bias that was a consequence of only five replicates in each of the four treatments.

Among predators the interaction between Fencing and Resource was more frequent and variable (i.e., the Resource effect occurred only in Open or Fenced plots with similar frequency). The more variable pattern is likely due to the lower abundances of predators compared to other trophic categories, which would lead to higher overall sampling variation.

In summary: Interactions of Fencing with the Resource treatment were present, but showed no consistent pattern for the less-abundant taxa, and for the more abundant microbi-detritivores, the consequence was to bring the overall evidence for bottom-up control closer to that of CW99.

Plot size — Plot area was 2.5 larger in CW99 than in our experiment (10 m2 versus 4 m2) but the effect of this difference on the possible swamping effect of migration is smaller than the 2.5 ratio might suggest, for three reasons: (1) the rectangular plot shape of CW99 (2 × 5 m) means that the perimeter-to-area ratio of our plots was only 1.4× greater than that of CW99 [(8/4) / (14/10); (2) microbi-detritivores showed a strong response to resource supplementation in open 1-m 2 plots in an earlier experiment in the same forest (Chen & Wise, 1997); and (3) the absence of consistently strong Fencing × Resource or Fencing × (Resource × Year) interactions suggests that migration across open plot boundaries did not strongly dilute or increase the responses of most taxa to effects of detrital supplementation.

Variation in rainfall— Biweekly rates of detrital supplementation were the same in Years 2 and 3, yet the patterns of response in both years differed markedly from each other. Furthermore, Year 3 patterns differed markedly from CW99, even though rates of detrital addition in the last two years of our experiment were close to those of CW99. One factor possibly contributing to these differences is year-to-year variation in rainfall, which during Year 2 of our experiment was close to normal but was ∼50% below normal during Year 3 ( Lawrence, 2000). During CW99 rainfall was 35% above the long-term average. (In Year 1 rainfall was close to normal ( Lawrence, 2000), which may explain why several taxa responded positively to the Resource treatment even though the rate of supplementation was only 1/3 that of CW99.) Higher-than-normal rainfall in the short-term experiment of CW99 may have accelerated fungal growth in the leaf litter, intensifying the effect of the detrital subsidy compared to all three years of our experiment. Furthermore, the much-lower rainfall in Year 3 could explain why densities of four Collembola families—Onychiuridae, Isotomidae, Tomoceridae and Sminthuridae—had declined to near zero in both Ambient and Supplemented plots by the fall of Year 3 (Figs. S5.2, S5.4–S5.6 in Appendix S5) (Christiansen, 1964; Hopkin, 1997; Petersen, 2002). In contrast to this pattern, however, both hypogastrurid and entomobryiid Collembola responded strongly and positively to Resource supplementation in the fall of Year 3, similarly to the fall of Year 2. Possibly these strong responses occurred despite the low rainfall because of release from competition with the other four Collembola families. Several long-term studies suggest that strong biotic interactions influence Collembola densities, particularly in forests (Chernova & Kuznetsova, 2000; Kampichler & Geissen, 2005; Kuznetsova, 2006; Takeda, 1987; Van Straalen, 1985; Vegter, 1987; Wolters, 1998). However, direct experimental evidence is lacking for resource competition among Collembola families or apparent competition due to shifting pressures from predation.

Comparison with other experiments testing for bottom-up control in terrestrial detritus-based food webs

Other experiments usually employed plot sizes comparable to, or smaller than, those we used, and they were conducted for less time. Two exceptions are grassland studies (Fountain et al. 2008; Patrick, Kershner & Fraser, 2012) in which inorganic nutrients were added for 4 years to very large (240 or 314 sq. m, respectively) unfenced plots. Fountain et al. (2008) found that densities of isotomid Collembola and three spider families increased, but densities of two other spider families decreased, in response to nutrient enhancement. In contrast, Patrick, Kershner & Fraser (2012) observed only positive responses by families of web-building and cursorial spiders. In another grassland study that used much smaller (1 × 1-m) open plots, Hoekman et al. (2011) added midge (Diptera: Chironomidae) carcasses for two years or over a single year starting at different times. As in our study, Collembola, larval Diptera, and Coleoptera exhibited varying positive responses. Spiders, however, showed no treatment effect, in contrast to our Year 1 and CW99. Because their plots were just 25% the area of our experimental units and 10% that of CW99, movement across plot boundaries could have diluted positive responses by spiders to increased prey. In addition, increased cannibalism and intraguild predation could have contributed to the absence of an effect on total spiders. Oelbermann, Langel & Scheu (2008) directly manipulated energy input to a grassland detrital web in fenced 5-m2 plots for five months. Many Collembola families responded positively as did total spider numbers, which were ca. 2 × higher in the detritus-enhanced plots. Some spider taxa, however, exhibited no numerical response even though stable-isotope analysis revealed that all sampled spiders relied more heavily on the decomposer food web in the detritus-addition plots. Oelbermann, Langel & Scheu (2008) suggested that even though the secondary-consumer trophic level was strongly linked to the detritus base, individual groups of predators showed inconsistent evidence of bottom-up control due to increased cannibalism and intraguild predation in the plots with additional detritus. Similar interactions could have weakened bottom-up control among predators in Year 1 (with delayed effects in Years 2 and 3), but one still has to explain the strong bottom-up control among predators in CW99.

The other comparable experiments on detritus-based systems have been conducted in forests. Scheu & Schaefer (1998) and Maraun et al. (2001) increased microbial growth in the litter layer of fenced 1-m2 plots by adding glucose and nitrogen for 15 months with no effect on Collembola or centipedes of the litter layer. In contrast, Collembola and centipedes of the lower soil horizon responded negatively, most likely due to indirect effects of increased earthworm densities (Scheu & Schaefer, 1998; Maraun et al., 2001). We did not sample lower soil layers and in our forest earthworms appeared to be much less abundant than in the German beech forest on limestone (Göttinger Wald) where these studies were conducted (D Wise, pers. obs., 1984). Also, centipedes were rare in our study, in contrast to the much higher abundance just a few years earlier in CW99—a difference for which there is no obvious explanation. One might speculate that the rarity of a major predatory group could have altered the pattern of bottom-up control among other predators, but a possible mechanism is illusive.

Another explanation for why responses observed by Scheu & Schaefer (1998) and Maraun et al. (2001) differed from ours is the quite different type of resource enhancement they employed. However, Salamon et al. (2006), who used a similar type of resource enhancement in 1-sq. m fenced plots for 17 months, found that several Collembola families did respond positively with effect sizes similar to those observed in our experiment. Unlike our Year 1, though, spiders did not respond, but the absence of a response by pseudoscorpions does resemble our findings.

The forest-floor experiment of Raub et al. (2014) is the closest to ours in technique, as every two weeks for 3 months they added to 1.5 × 1.5-m unfenced plots the same type of artificial detritus used in CW99 and our study. Collembola numbers responded positively to resource enhancement, but total predators (spiders, pseudoscorpions, and centipedes combined) did not increase.

It is difficult to discern a pattern from such disparate studies, except that the general absence of bottom-up control among predators is similar to our results for Years 2 and 3 (but not for Year 1 nor CW99). Many studies did not employ the same degree of taxonomic resolution employed by us and CW99, which means that the absence of a response for an entire grouping (i.e., Collembola, predators) does not imply that smaller taxonomic categories did not respond positively and/or negatively. We avoided this problem by first using a multivariate approach to examine changes in community structure for a reasonably large number of response variables. Variation between studies in abiotic factors, such as rainfall, could explain some of the differences in outcomes, but that data is generally lacking, or the experiment was usually not conducted for a long enough time for rainfall differences to appear. The openness of the plots, especially for those that were smaller than those in CW99, could explain some of the variable responses. Our 3-year experiment, with open and fenced plots and the ability to compare directly with CW99, overcomes many of these challenges to interpretation.

Towards a deeper understanding of bottom-up control in detrital food webs

Some of the weakening of bottom-up control between Years 2 and 3 in our experiment can be explained by 50% less rainfall in Year 3. In detrital food webs a weak positive response by microbi-detritivores to resource addition is expected if drought impedes fungal growth. However, lower rainfall, although a likely contributing factor, cannot entirely explain the weaker bottom-up control in Year 3. Fungal hyphae were still 3× more abundant in Supplemented than Ambient plots in Year 3 (Appendix S1), and the total amount of detrital material (leaf litter plus added detritus) at the end of Year 3 was only 13% greater in Supplemented than Ambient plots, compared to a ∼30% difference for CW99, when rainfall was higher than any year of our experiment. Thus, fungal growth may still have been substantial in Year 3. Although drier conditions in the litter could explain why densities of four Collembola families were close to zero in all plots in the fall (and hence failed to exhibit bottom-up control), not all primary consumers exhibited a fall crash in Year 3 (Appendix S5). In addition, examination of changes in densities of Collembola families during the summer reveals no correlation with changes in rainfall over the three years of the experiment. Most critically, declining rainfall cannot readily explain the negative responses to detrital supplementation. Why was one major Collembola family, the Tomoceridae, less abundant in Supplementation than Ambient plots in Year 3? In this forest predation by cursorial spiders depresses tomocerid densities (Wise, 2004), but cursorial spiders were less, not more, abundant in the Supplementation treatment in Year 3. In fact, the widespread negative responses of predators to detrital addition the last two years was surprising when compared with CW99 and other studies. It is unclear how all these negative responses could result from shifting intensities of intraguild predation and cannibalism in response to increased abundances of microbi-detritivores. Possibly contributing factors are declining numbers of high-quality prey (Tomoceridae) (Toft & Wise, 1999b) and increased densities of possibly toxic isotomid (Toft & Wise, 1999a; Toft & Wise, 1999b) and hypogastrurid prey (Messer et al., 2000; Negri, 2004) in Years 2 and/or 3 in the Supplemented plots. It is difficult to evaluate the solidity of such post-hoc theorizing without additional information.

In addition to the appearance in the last two years of negative responses to detrital supplementation by predators and tomocerids, variation in community structure between Supplemented plots was greater than between Ambient plots the fall of Year 3—the only sampling period that displayed this difference (Fig. 2). This increased variation in community structure after three years, and all the unexpected contrasts of the univariate analyses with CW99, suggest that that long-term resource supplementation will lead to a community configuration that cannot be predicted from initial short-term population responses. On the other hand, could such negative responses in long-term experiments simply be random vagaries emerging from the low replication in most field experiments? Or might they be the surprising results that ecologists often fail to report (Doak et al., 2008)—the unexpected observations that can help build stronger theory? Additional long-term experiments will help resolve this dilemma.

How long is long enough? The behavior of mathematical models of long-term press perturbations suggests that many generations will be required to reach a new equilibrium (Bender, Case & Gilpin, 1984; Raffaelli & Moller, 2000; Yodzis, 1988), which coupled with environmental noise, such as variation in rainfall, might make the prospects of a “long-enough” field experiment seem hopeless. However, after reviewing short- and long-term experiments (2–31 months) in the intertidal, Menge (1997) concluded that “. . . community dynamics may be more predictable than expected . . .” because most indirect effects had appeared half-way through the experiment. “Long enough” is at least the number of generations sufficient to reveal indirect effects. An even longer time would reveal how close the community may be to a new equilibrium. This longer time will never be achieved with certainty, but snapshots yielded by short-term experiments cannot answer the question.

Simply advocating experiments of longer duration is not enough. Even in long-term experiments we need to reduce the impact of unpredictable and unknown variation in other factors. Furthermore, we must have a deeper understanding of which interaction pathways were altered in response to resource supplementation. How might these two challenges be met?

The challenge of accounting for uncontrollable yearly variation in abiotic and biotic factors can be addressed by expanding the experimental design to include initiation of resource additions in new sets of experimental units in each subsequent year after the experiment has started, along with continuing the experimental treatment in the plots from the first year. Phased temporal replication of the experimental perturbation offers the surest way of clearly separating effects due to time lags from those due to changing levels of unknown but influential abiotic and biotic factors. Hoekman et al. (2011) employed a preliminary but elegant version of this approach by combining two yearly pulse treatments of detritus addition with a two-year press treatment.

The second challenge can be met in several ways. One is the use of path analysis and structural equation modeling to help separate direct and indirect effects among interaction pathways between taxa in major trophic groupings in the food web (Grace, 2006; Wootton, 1994). However, this statistical modeling approach requires numerous replicates, more than are possible with most types of detrital enhancement experiments performed to date. Tradeoffs between too-many response variables, too-few replicates, and experimental units too small to be realistic present daunting challenges. Another approach is to measure more variables than just the densities of primary consumers and their predators. Molecular techniques are available to measure different functional categories of fungi (Nguyen et al., 2016; Shokralla et al., 2012). Shifts in major trophic pathways can be revealed by techniques such as stable isotopes and fatty acid analysis (Halaj & Wise, 2002; Ruess et al., 2004), and PCR of prey DNA in guts of predators such as spiders and carabids (Harper et al., 2005; Symondson, 2002).

Some of the above suggestions already have been incorporated into short-term field experiments testing for bottom-up control in detritus-based terrestrial food webs. Utilizing all of them in a single study is a challenge. Nevertheless, paying heed to their advantages when planning future long-term research would help reveal likely causes of the type of variability found in our 3-year experiment.

Supplemental Information

Appendix S1 Analysis of ergosterol in leaf litter

Click here for additional data file.

Appendix S2 Methodological details of the statistical analyses

Click here for additional data file.

Appendix S3 Multivariate analyses with perMANOVA: interactions of Resource × Year with Season

Click here for additional data file.

Appendix S4 Multivariate analyses (perMANOVA) of the Resource effect for each Year/Season combination

Click here for additional data file.

Appendix S5 Univariate Plots: patterns of change over time of the individual response variables, Figs S5.1–S5.18

Click here for additional data file.

Appendix S6 Results of univariate analyses and patterns of CAP vector overlays, Tables S6.1–S6.4

Click here for additional data file.

Appendix S7 Arthropod data set

Click here for additional data file.

Appendix S8 Raw data

Click here for additional data file.

We thank Berea College for permission to conduct the research in the Berea College Forest. Keith Erny, Sara Todd, Meredith Houston, Tom Coleman, and Adam Keener provided field and laboratory assistance; and Lowell Bush and Neil Fannin provided valuable help with use of HPLC to assay for ergosterol. Suggestions of Robin Mores, Matthew McCary, Monica Farfan, Crystal Guzmán, Christian Mulder, Daniel Grunder, Shaun Turney, and anonymous reviewers greatly improved the manuscript.

Additional Information and Declarations

Competing Interests

Author Contributions

Data Availability

Dr. Lawrence is a personal consultant, and the sole proprietor of KL2 Consulting, LLC. The authors declare that they have no competing interests.

Kendra L. Lawrence conceived and designed the experiments, performed the experiments, contributed reagents/materials/analysis tools, wrote the paper, reviewed drafts of the paper.

David H. Wise conceived and designed the experiments, analyzed the data, contributed reagents/materials/analysis tools, wrote the paper, prepared figures and/or tables, reviewed drafts of the paper.

The following information was supplied regarding data availability:

The raw data has been supplied as a Supplementary File.

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
