# Peer review of "Long-term resource addition to a detrital food web yields a pattern of responses more complex than pervasive bottom-up control"

_PeerJ, doi:10.7717/peerj.3572_

## Round 0.1 · original submission · Major Revisions

Thank you for your revised MS/rebuttal, and thanks to the reviewers for their work on this. One reviewer suggested minor revisions, the other major. In the case of the major revision recommendation, none of the requested changes require further experimentation, but do require a response regarding design and statistical methodology.

I also note that reviewer 2 states that:

"I feel this work constitutes an important contribution to the literature and will, hopefully, stimulate further work investigating these ideas."

...and reviewer 1 states that:

"I feel this manuscript is much improved! It is well-written and the subject is interesting and relevant."

I tend to agree with these assessments, so while I am giving this a major revisions recommendation – in order to remain on the conservative side of the decision fence – I will assess the revised MS and rebuttal by the authors in terms of whether it needs to go back to reviewer 2 for one more look at that time.

Thanks again to all of you for your work and professionalism in this review process.

·

Basic reporting

The manuscript is well-written and easy to follow. I'm sorry my initial comment about the punctuation was unclear. (Or if it implied the authors are not strong writers - I only meant that having an outside eye can help to catch small issues.) I was indeed referring to the heavy use of commas and semi-colons. Now that they've been reduced, I find the manuscript reads more smoothly.

Two small issues, in my opinion, remain:

(1) I'm still not sure I'm on board with Figure 4. I definitely prefer the new caption. However I still feel that representing both effect size and p-value using the same symbol can't really be justified.

(2) I stand by my previous comments and the comments of Review 2 about the discussion. The description of previous experiments without relating back to your results (552-591) is not what one usually expects to find in a discussion.

Experimental design

I feel your reply to my comments and your edits do a good job of addressing my criticism of your experimental design. I especially like that the fencing treatment has been re-framed to be more about the fencing itself, rather than dispersal.

Validity of the findings

As you have pointed out in your reply to the reviewers and editor, your statistical philosophy diverges somewhat from the status quo. I feel that is totally fine as long as you're clear about your statistical methods and include your raw data, which you've done.

I agree with you that identifying specimens to the level of order is adequate for the questions being posed, especially in the context of a comparison to CW99.

Additional comments

I feel this manuscript is much improved! It is well-written and the subject is interesting and relevant.

One last small note: in the acknowledgements section, my name is "Shaun", not "Shawn"! :)

Reviewer 2 ·

Basic reporting

no comment

Experimental design

1. I see no problem with presenting and discussing Year 1 data. However, time is confounded with supplementation rate for [Year 1] vs [2 & 3], so I’m not comfortable with the inclusion of Year 1 data in analyses calculating interaction terms for Time and Fencing. Why not assess the time and fencing interaction terms using only year 2 & 3 data? I see the authors have calculated the Time interaction term using only years 2 & 3 in a Supp Mat, but I feel Years 2 & 3 should comprise the basis for all analyses and inferences drawn from the interaction terms. Doing so would not detract from the experiment’s ability to assess whether the response to supplementation changes over time.

2. Investigating the effect of fencing is stated as a central goal of this manuscript. I do question the ability of this experiment to detect effects of fencing with respect to the points below - but I do not object to publication on these grounds, as long as the authors are abundantly clear about these limitations:

(i) capacity of fencing to prevent movement, due to the enclosures’ open top design.

If the authors state they test for effects of fencing, I take some issue with the statement that ‘ecologists use fencing to impede movement’. I encourage publication if the authors are more explicit and upfront about how much migration their fencing treatment can be truly expected to prevent. The text in lines 72-83 does not fulfill this need.

If fencing was not a good barrier to movement, surely there would be greater variation in fenced vs open plots for univariate analyses at the functional group level – the authors may wish to comment on this.

(ii) statistical power (five replicates each of open / fenced within each supplementation treatment).

Providing a power analysis for this aspect of the experiment (instead of discussing the experiment’s overall power, line 584) would provide more suitable justification and context for their use of p = 0.15. I see no issue with using p=0.15 per se.

Validity of the findings

3. Fig. 4: I do find that adjusting effect size for p-value is a bit odd and obscures the results. Readers should not have to pore over the Supp Mat tables to get the full picture. Why not reserve arrow thickness to indicate absolute effect size and simply add a number of stars beside each arrow to indicate significance level?

Additional comments

4. I commend the authors for endeavoring to replicate the well-cited Chen & Wise 1999 study. This study was important in establishing bottom-up limitation of multiple trophic levels in detritus-based food webs, and is often cited as a key example documenting a clear effect. I see a twofold value of this manuscript: (i) it attempts to replicate the above work, and (ii) it establishes and attempts to test hypotheses for variation in published experimental findings. I feel this work constitutes an important contribution to the literature and will, hopefully, stimulate further work investigating these ideas.

5. Comments about the food web aspect of this work:

Line 444, “particularly in Year 1”: I do not see this in Fig. 4.

I see a few interesting aspects of the results at the gross community level; the authors may wish to mention these, at their discretion:

Lines 149-150: I don’t recall seeing the authors comment on this in the discussion. This is interesting because it suggests that significant detrital consumption occurred in supplemented plots, despite apparently few responses across taxonomic groups by the end of Year 3.

The strongest effects, according to the current method of establishing effect size in Fig 4, manifested among a few of the most numerically dominant groups. This further supports bottom-up limitation of primary consumers at a gross community level.

Again judging from Fig 4: Among taxa that were evaluated by Chen & Wise 1999, results are largely replicated among primary consumers and the mixed trophic level in year 2 data, the first year with comparable supplementation rate to CW99. In the final year, biomass then seems to accumulate in a few numerically-dominant primary consumers. It would be useful to those studying food webs to know whether there occurred a shift in relative abundance towards these taxa. If this is the case, then the overall result at the gross community level remains unchanged between years 2 & 3, and there is simply a reduction in diversity.

If the authors’ rainfall hypothesis is correct, densities of strict fungivores (or primary consumers capable of consuming fungal hyphae) should have risen in years of high rainfall – was this the case?

6. Minor comments:
Lines 74-83: could be removed, significantly reduced, or moved to the discussion.

Line 385: I believe “Year 3” should be “Year 2”

Line 428: I believe “Fig. S4.4” should be “Fig. S5.4”

Line 718: the meaning of this sentence is not clear to me

---

## Round 0.2 · accepted · Accept

Thank you to the reviewers for excellent suggestions, and to the MS co-authors who took up every one of the reviewers' suggestions (including the statistical suggestions). This MS has definitely been strengthened through concerted efforts by all involved and is now suitable for publication in PeerJ.

---

## Author Rebuttal · Round 0.2

Institute for Environmental Science and Policy (MC 673)
529 School of Public Health West
2121 West Taylor Street
Chicago, Illinois 60612-7260

10 June 2017

Dr. Dezene Huber
Academic Editor, PeerJ

Dear Dezene,

Kendra Lawrence and I pleased to submit an extensively revised version of our manuscript "*Long-term resource addition to a detrital food web yields a pattern of responses more complex than pervasive bottom-up control.*"

I apologize for the delay in sending you our revision. In addition to outside forces eating up my hours, a major cause of our tardiness is our decision to respond to all the reviewers' comments, which required a complete re-analysis of the data and a major re-write of the Results and Discussion, with more minor changes to the Methods and Abstract.

We applaud the stubbornness of both reviewers in sticking to their positions, and are especially grateful for the thoroughness and thoughtfulness of Reviewer 2. We have made all the requested changes, even those suggested by Reviewer 2 as being optional. Thus, we have no rebuttal sections.

We re-analyzed the data following the suggestions of Reviewer 2. This new approach meant that we did not have to use permANOVA for all our univariate statistical modelling, but could employ GLM for Year 1 and GLMM for Years 2 and 3 for a majority of univariate analyses. The consequences of the fact that more data now fit the generalized linear model is a much clearer pattern in the results, which is reflected in our completely revised Figure 4. In response to suggestions by both reviewers, in this new version arrow width solely reflects effect size, and strength of evidence is indicated by asterisks related to the *P* value of the term from the statistical model.

The new statistical approach resulted in major revisions of Appendices S2, S3, S4 and S6 – and a complete revision of S5. The complexity of the old S5 was confusing; it has been replaced with plots of all 18 response variables, which will be much more useful to readers who wish to examine the patterns in more detail. Appendix S6, which has been expanded from two to four tables, is now much easier to digest. Appendix S6 now summarizes simple Resource effects for Year 1, and the Resource x Year interaction for Years 2 and 3, following the suggested change in statistical modeling recommended by Reviewer 2.

Our overall conclusions are the same, hence the title has not been changed and the Abstract has been tweaked just a little. The major consequence of the new analysis is that the patterns are much clearer and the Discussion is more focused at the same time that the implications of our findings can be placed in a broader context than before.

Below we have pasted our explanations of how we responded to the suggestions of both reviewers. We have not indicated line numbers for our changes because the manuscript has been so extensively revised. However, we have included a file with "Track Changes" for comparison with our revision. In meeting the reviewers' recommendations, we had to expand several portions of the manuscript; however, we were able to cut other sections, so that this revision is close to the length of the original submission.

On a personal note, I must confess that at the beginning I was frustrated with all the time that this revision was taking, but we are now very much happier with the manuscript, and are convinced that your suggestions and those of both reviewers have improved the manuscript immensely.

Warm regards,

David H. Wise
Associate Director, Institute for Environmental Science and Policy
&
Professor of Ecology and Evolution, Department of Biological Sciences

Our responses to the reviewers are ***written in bold italics and underlined.***

**REVIEWS OF JANUARY 2017 SUBMISSION TO PeerJ:**

Dear David,
Thank you for your submission to PeerJ.
It is my opinion as the Academic Editor for your article - Long-term resource addition to a detrital food web yields a pattern of responses more complex than pervasive bottom-up control - that it requires a number of Major Revisions.
My suggested changes and reviewer comments are shown below and on your article 'Overview' screen. If you address these changes and resubmit, there's a good chance your article will be accepted (although this isn't guaranteed).

Resubmission Checklist
Download to begin the resubmission process
Download Checklist
Although not a hard deadline, we expect you to submit your revision within the next 55 days.

With kind regards,
Dezene Huber
Academic Editor, PeerJ

**Editor's Comments**
MAJOR REVISIONS
Thank you for your revised MS/rebuttal, and thanks to the reviewers for their work on this. One reviewer suggested minor revisions, the other major. In the case of the major revision recommendation, none of the requested changes require further experimentation, but do require a response regarding design and statistical methodology.

I also note that reviewer 2 states that:

"I feel this work constitutes an important contribution to the literature and will, hopefully, stimulate further work investigating these ideas."

...and reviewer 1 states that:

"I feel this manuscript is much improved! It is well-written and the subject is interesting and relevant."

I tend to agree with these assessments, so while I am giving this a major revisions recommendation – in order to remain on the conservative side of the decision fence – I will assess the revised MS and rebuttal by the authors in terms of whether it needs to go back to reviewer 2 for one more look at that time.
> ***We very much appreciate this recommendation and agree that the required revisions are sufficiently substantial that a final decision must await an evaluation of how successfully we have satisfied the reviewers' concerns.***

Thanks again to all of you for your work and professionalism in this review process.

**Reviewer 1 (Shaun Turney)**
Basic reporting
The manuscript is well-written and easy to follow. I'm sorry my initial comment about the punctuation was unclear. (Or if it implied the authors are not strong writers - I only meant that having an outside eye can help to catch small issues.) I was indeed referring to the heavy use of commas and semi-colons. Now that they've been reduced, I find the manuscript reads more smoothly.

Two small issues, in my opinion, remain:

(1) I'm still not sure I'm on board with Figure 4. I definitely prefer the new caption. However I still feel that representing both effect size and p-value using the same symbol can't really be justified.
> ***We agree. Figure 4 has been completely revised. Arrow width now indicates effect size, and asterisks next to the arrow indicate the P value of the effect from the statistical model.***

(2) I stand by my previous comments and the comments of Review 2 about the discussion. The description of previous experiments without relating back to your results (552-591) is not what one usually expects to find in a discussion.
> ***This section has been completely revised so that the design and results of other studies now relate to the design and findings of our experiment.***

Experimental design
I feel your reply to my comments and your edits do a good job of addressing my criticism of your experimental design. I especially like that the fencing treatment has been re-framed to be more about the fencing itself, rather than dispersal.

Validity of the findings
As you have pointed out in your reply to the reviewers and editor, your statistical philosophy diverges somewhat from the status quo. I feel that is totally fine as long as you're clear about your statistical methods and include your raw data, which you've done.

I agree with you that identifying specimens to the level of order is adequate for the questions being posed, especially in the context of a comparison to CW99.

Comments for the Author
I feel this manuscript is much improved! It is well-written and the subject is interesting and relevant.

One last small note: in the acknowledgements section, my name is "Shaun", not "Shawn"! :)
>      ***OK, Done!***

**Reviewer 2 (Anonymous)**
Basic reporting
no comment

Experimental design
1. I see no problem with presenting and discussing Year 1 data. However, time is confounded with supplementation rate for [Year 1] vs [2 & 3], so I'm not comfortable with the inclusion of Year 1 data in analyses calculating interaction terms for Time and Fencing. Why not assess the time and fencing interaction terms using only year 2 & 3 data? I see the authors have calculated the Time interaction term using only years 2 & 3 in a Supp Mat, but I feel Years 2 & 3 should comprise the basis for all analyses and inferences drawn from the interaction terms. Doing so would not detract from the experiment's ability to assess whether the response to supplementation changes over time.
>      ***Excellent Point. We agree completely. All the statistical analyses (multivariate and univariate) have been re-done. For Year 1 we have used a 2-way model (Resource x Fencing), for the remaining two years a 3-way model (Resource x Year x Fencing). Thus we now test for the interaction with year only for Years 2 and 3. We have changed the text, Figure 4, and the Supplemental Appendices (S2, S3, S4 and S6) accordingly. This new approach not only parallels the experimental design much better, it results in our being able to use generalized linear models for the majority of the analyses, which was not true when we included all three years in the model. The results are now much clearer.***

2. Investigating the effect of fencing is stated as a central goal of this manuscript. I do question the ability of this experiment to detect effects of fencing with respect to the points below - but I do not object to publication on these grounds, as long as the authors are abundantly clear about these limitations:

(i) capacity of fencing to prevent movement, due to the enclosures' open top design.

If the authors state they test for effects of fencing, I take some issue with the statement that 'ecologists use fencing to impede movement'. I encourage publication if the authors are more explicit and upfront about how much migration their fencing treatment can be truly expected to prevent. The text in lines 72-83 does not fulfill this need.

> ***OK, Done. We have removed these lines and have revised the manuscript to clarify that fencing only impedes movements of some, not all groups; and we have expanded the discussion points regarding the influence of fencing on response patterns of different taxa. Fencing is routinely used to impede movements of epigeic (ground active arthropods) in these types of field experiments, although open plots are often used also. We could cite an extensive literature, but that would take up too much space. In the Discussion it is now made clear that studies of this type employ either open or fenced plots. Our study is the first to evaluate the impact of fencing.***

If fencing was not a good barrier to movement, surely there would be greater variation in fenced vs open plots for univariate analyses at the functional group level – the authors may wish to comment on this.

> ***?? Don't understand this argument . . .should the sentence read "If fencing was a good barrier to movement, surely . . . .. "?? This may be what the reviewer meant, and it is a good point, but because in our revision we now discuss the impact of fencing in more detail than before, we don't want to lengthen our current discussion of fencing because we think it will detract from the major points of the paper. Anyone interested in examining the impact of fencing on the SE of population size of different groups can examine the plots of all the response variables in the new Appendix S5.***

(ii) statistical power (five replicates each of open / fenced within each supplementation treatment).

Providing a power analysis for this aspect of the experiment (instead of discussing the experiment's overall power, line 584) would provide more suitable justification and context for their use of p = 0.15. I see no issue with using p=0.15 per se.

> ***We agree that this point needed clarification. We have expanded considerably our discussion of statistical power with respect to the number of replicates in each treatment, and the impact of the Fencing treatment on the power to detect Resource effects with respect to the statistical power of Chen and Wise 1999 (CW99). In fact, we now have a section in the Discussion devoted to discussing the impact of including a Fencing treatment on the power to detect a Resource effect.***
>
> ***In the revision we now use P = .10 as a cutoff for examing an interaction term or reporting an effect in a statistical model. This is not controversial in the research community so we have not included a power analysis to justify it – which would be difficult anyway given the number of different models we now use for the univariate modeling.***

Validity of the findings
3. Fig. 4: I do find that adjusting effect size for p-value is a bit odd and obscures the results. Readers should not have to pore over the Supp Mat tables to get the full picture. Why not reserve arrow thickness to indicate absolute effect size and simply add a number of stars beside each arrow to indicate significance level?

> ***We now agree completely. Figure 4 has been extensively revised. Arrow width now indicates effect size, and asterisks next to the arrow indicate the P value of the effect from the statistical model.***

Comments for the Author

4. I commend the authors for endeavoring to replicate the well-cited Chen & Wise 1999 study. This study was important in establishing bottom-up limitation of multiple trophic levels in detritus-based food webs, and is often cited as a key example documenting a clear effect. I see a twofold value of this manuscript: (i) it attempts to replicate the above work, and (ii) it establishes and attempts to test hypotheses for variation in published experimental findings. I feel this work constitutes an important contribution to the literature and will, hopefully, stimulate further work investigating these ideas.

5. Comments about the food web aspect of this work:

Line 444, "particularly in Year 1": I do not see this in Fig. 4.
> ***Agree, excellent point. This paragraph has been rewritten.***

I see a few interesting aspects of the results at the gross community level; the authors may wish to mention these, at their discretion:

Lines 149-150: I don't recall seeing the authors comment on this in the discussion. This is interesting because it suggests that significant detrital consumption occurred in supplemented plots, despite apparently few responses across taxonomic groups by the end of Year 3.
> ***Thanks, this is an excellent point. It is now incorporated into a new paragraph in the Discussion in which we discuss how much of an effect lower rainfall may have had on population responses of fungivores/detritivores.***

The strongest effects, according to the current method of establishing effect size in Fig 4, manifested among a few of the most numerically dominant groups. This further supports bottom-up limitation of primary consumers at a gross community level.

Again judging from Fig 4: Among taxa that were evaluated by Chen & Wise 1999, results are largely replicated among primary consumers and the mixed trophic level in year 2 data, the first year with comparable supplementation rate to CW99.
> ***We agree. The revised analyses makes this pattern clearer. In the revised Discussion we now emphasize the above point about the Year 2 responses, and also give more emphasis to the disappearance of bottom-up control (and appearance of NEGATIVE effects) among predators in Years 2 and 3. Bottom-up control among predators was there, but weak, in Year 1 despite the lower rate of detrital supplementation that year; the revised statistical analysis that you suggested has revealed that evidence for bottom-up control for all trophic levels was stronger for Year 1 than was apparent from our flawed analysis in which we included all three years for the Resource x Year interaction.***

In the final year, biomass then seems to accumulate in a few numerically-dominant primary consumers. It would be useful to those studying food webs to know whether there occurred a shift in relative abundance towards these taxa. If this is the case, then the overall result at the gross community level remains unchanged between years 2 & 3, and there is simply a reduction in diversity.

*Excellent point – although by fall many primary consumers were much less abundant in all three years (a common phenological pattern), this didn't occur in the Summer samples, not even in Year 3. We have addressed the issue of possible effects of rainfall on population sizes in greater detail in the revision (see responses above and below), including the fact that examining summer values across all three years shows no clear correlation with changes in rainfall. The reader can now examine these changes in abundance for all variables for both seasons by examining the new plots in Supplemental Appendix S5.*

If the authors' rainfall hypothesis is correct, densities of strict fungivores (or primary consumers capable of consuming fungal hyphae) should have risen in years of high rainfall – was this the case?

    *Excellent point; in the Discussion we now treat the issue of declining populations of some primary consumers in Year 3 in much more detail; as a consequence, we have tempered our inferences about the possible impact of changing rainfall (see comment above).*

6. Minor comments:

Lines 74-83: could be removed, significantly reduced, or moved to the discussion.
    *Agree; removed.*

Line 385: I believe "Year 3" should be "Year 2"
    *Yes; changed.*

Line 428: I believe "Fig. S4.4" should be "Fig. S5.4"
    *Yes, but Appendix S5 has been completely changed to this typo is no longer relevant.*

Line 718: the meaning of this sentence is not clear to me
    *Introductory sentences in this paragraph have been modified so that hopefully their meaning is clearer.*

Article ID: 15568